# Risky Sexual Behaviour and Sexual Victimization among Young People with Risky Substance Use in Europe—Perspectives from Belgium, Sweden, the Czech Republic, and Germany

**DOI:** 10.3390/ijerph20217002

**Published:** 2023-10-31

**Authors:** Christiane Baldus, Tobias H. Elgán, Veerle Soyez, Hanne Tønnesen, Nicolas Arnaud, Ladislav Csemy, Rainer Thomasius

**Affiliations:** 1German Centre for Addiction Research in Childhood and Adolescence (DZSKJ), University Medical Centre Hamburg-Eppendorf, Martinistrasse 52, 20246 Hamburg, Germany; 2STAD, Centre for Psychiatry Research, Department of Clinical Neuroscience, Karolinska Institutet & Stockholm Health Care Services, Region Stockholm, 11364 Stockholm, Sweden; tobias.elgan@ki.se; 3Department of Psychology, Vrije Universiteit Brussel, 1050 Brussels, Belgium; veerle.soyez@vub.be; 4Clinical Health Promotion Centre, Lund University, 22185 Lund, Sweden; 5Public Mental Health Research Program, National Institute of Mental Health, 25067 Klecany, Czech Republic

**Keywords:** adolescence, binge drinking, illicit drug use, sexual victimization, sexual risks, condom use

## Abstract

Background: Research indicates that among the risks associated with young people’s alcohol and illicit drug use are sexual risks. However, insights into co-occurrence of substance use and sexual risks in adolescent samples and possible differences across countries are limited. Methods: A sample of 1449 adolescents from Belgium, Sweden, the Czech Republic, and Germany screened positive for risky alcohol/illicit drug use in a web-based intervention against alcohol and illicit drug use. They also reported incidents of sex while being drunk and/or high on drugs, condomless sex on these occasions, and sexualized touching and sexual victimization while being drunk or high on drugs. Results: In the sample, 21.5% of the participants reported sexualized touching, 9.9% being victim to sexual assault, and 49.8% having had sex while being drunk and/or high on drugs; of the latter, 48.3% had condomless sex. Reports on having had sex while being drunk and/or high on drugs were associated with higher levels of past 30-day binge drinking. Being a victim of sexual assault was associated with past 30-day binge drinking only in young men. Conclusion: When devising preventive interventions against risky substance use in adolescents, an additional focus should be set on integrating steps against sexual risks.

## 1. Introduction

Alcohol and illicit drug use among adolescents is widespread and a major public health concern [1,2]. Adolescent heavy episodic drinking is associated with a number of unwanted outcomes such as lower academic achievement, smoking [3], and violent attacks [4]. Among the risks further mentioned in association with adolescent substance use are sexual risks, which include risky sexual behaviour [5] and reported sexual victimization [6]. This is especially noteworthy because early sexual experiences shape future sexual risks and health behaviour [7]. The aims of the current paper are first, to examine levels of sexual risk, including sex while drunk and/or high on drugs, condomless sex, reported sexualised touching victimization, and reported sexual assault victimization, all of which occurred while subjects were drunk and/or high on drugs, among adolescents who had screened positive for risky alcohol/drug use; second, to explore how recent binge drinking or illicit drug use may increase the likelihood of reporting sexual risks; and third, to do so using a gender- and country-sensitive approach. The gender-sensitive approach is important because theory and research have pointed to gender differences in sexual risks. The country-sensitive approach is important because the data on substance use and sexual behaviour among young people in the countries included in this study, namely Sweden, the Czech Republic, Belgium, and Germany, differ. We take this as an indication of our assumption that differences may apply to the interplay between substance use and sexual risks investigated in the present research. Ultimately, our research aims to investigate whether it is worthwhile extending prevention efforts against risky substance use to include health-related messages about functional sexual behaviour. 

### 1.1. Risky Sexual Behaviour

The concept of risky sexual behaviour focuses on three unwanted outcomes: sexually transmitted infections (STIs), early or unwanted pregnancies, and the inability to choose sexual partners [8]. Risky sexual behaviours associated with these outcomes include having sex while being drunk and/or high on drugs [5,9] and condomless sex while being drunk and/or high on drugs [5,10]. Previous studies reporting data on sexual contact with concurrent alcohol and/or drug use vary widely in the samples used and mostly refer to college students [10] or young adults [5]. Data on adolescents are scarce as well as information on how risky sexual behaviour might vary across different countries. 

### 1.2. Sexual Victimization

Another aspect of sexual health involves unwanted or coerced sexual contact [11]. This comprises unwanted sexualized touching victimization [12] or sexual assault victimization. Taking advantage of someone being drunk or making someone drunk to engage in sexual activity are among the strategies of sexual coercion [13]. Sexual assault is associated with high risks for physical and mental health problems [14,15]. 

According to common belief, there are consistent gender differences in sexual victimization and virtually all data on sexual victimization are therefore reported separately by gender. Feminism underlines the role of power imbalance between males and females as a driving force towards female sexual victimization [16]. In fact, gender differences are widely reported in sexual victimization. Most studies examining sexual victimization do so regardless of victims’ concurrent alcohol and/or drug use: Landstedt and Gådin [15] found adolescent females to be more than twice as likely to have experienced sexual assault in Swedish 17-year-old students (females: 17.3%; males: 6.4%) than adolescent males. De Graaf and colleagues [17] reported lifetime prevalence rates for sexual assault at 4.2% in young adult men and 17.8% in women. In one of the few studies that reported data on sexual victimization while victims were drunk and/or high on drugs, Snipes and colleagues [12] found 5.1% of male college students and 12.1% of female college students reporting sexual victimization in which they were incapacitated by alcohol and/or drug use. Similar data on adolescent samples as well as cross-country comparisons of levels of sexual victimization while being drunk and/or high on drugs are widely missing.

### 1.3. Risky Sexual Behaviour, Sexual Victimization, and Alcohol and/or Illicit Drug Use

Both risky sexual behaviour and sexual victimization are conceptually linked to substance use, and the use of alcohol and illicit drugs is widely recognized as playing a major role in experiences of risky sexual behaviour and sexual victimization (e.g., [18]). Yet, different models exist with regard to the nature of these associations. On a situational level, risky sexual behaviour and sexual victimization often happen in party settings which often involve alcohol and illicit drug use [18,19]. Alcohol use may impair executive functioning, boost risky decision making [20], or leave heavy drinkers entirely incapacitated. Shared factors such as low levels of risk perception [21] or reward-sensitivity [22] may further contribute to the association between risky sexual behaviour and alcohol and drug use. Alcohol use and sexual victimization may both be influenced by engaging with deviant or older peers [23] because young people wish to display seemingly “mature” behaviour [24], early pubertal timing [25], and lower levels of parental monitoring [21]. 

Evidence suggests an association between participants’ levels of alcohol and illicit drug use and risky sexual behaviour. Frequency of past week binge drinking and past month illicit drug use were both associated with condomless sex among young people [11]. Discussions on how to explain links between risky sexual behaviour and past week or past month use patterns in particular have circled around the idea that these measures serve as a proxy to habitual use [26]. Moreover, among young people who had reported higher levels of alcohol use in previous years or higher levels of current usual alcohol use, higher frequencies of being drunk the last time they engaged in sexual behaviour [26], higher levels of unprotected sex while being drunk [5], and higher levels of risky sexual behaviour were found [26]. In the few event-level studies, which focus on situations to which respondents report on condom use or unprotected sex while being drunk and/or high on drugs, associations between alcohol or drug use and condom use were inconsistent [10]. 

With regards to gender, previous research has shown few gender differences in adolescents’ risky sexual behaviour and concurrent alcohol or drug use [11,26,27], although Kiene and colleagues [28] reported greater probabilities for condomless sex with unsteady partners when young women reported having used more alcohol in respective situations, while this association was less pronounced in young men.

Evidence on the association between sexual victimization and alcohol or illicit drug use often builds on global association studies [13,29]. Miller and colleagues [3] found high school students with a history of binge drinking to be more inclined to be sexually active or to report being victim of sexual assault. Kuttler and colleagues [6] found elevated prevalence rates of lifetime sexual victimization among adolescents with a history of alcohol-related emergency treatment. Concepts about linking sexual victimization and alcohol or illicit drug use mention a possible traumatogenic effect of sexual victimization and using substances to cope with the subsequent symptoms [30]. 

Only very few studies relate gender to the association between substance use and sexual victimization. Among the few studies relating substance use to sexual victimization while being incapacitated due to alcohol and/or drugs, a positive small correlation between past month use of alcohol mixed with energy drinks and sexual victimization in male college students (r = 0.13), but no such correlation in female college students (r = 0.02) [12]. It is not known whether gender is a moderator in the relationship between alcohol and/or drug use and sexual victimization as indicated by Snipes and colleagues [12] and whether these relationships vary across countries. Data on younger samples and the role of illicit drug use are scarce. 

### 1.4. Alcohol and Illicit Drug Use and Sexual Risks in Different European Countries

Both sexual behaviour and alcohol and illicit drug use are heavily influenced by cultural and societal context [31,32]. As the current study was undertaken in four different European countries—Sweden, Germany, the Czech Republic, and Belgium—previous data on adolescents’ sexual risks and alcohol and illicit drug use that allow cross-country comparisons are of interest. Among 10th graders in Europe, past 30-day drunkenness was reported by 21% of participants in the Czech Republic, 21% in Germany, 14% in Sweden, and 12% in Belgium. Gender differences were significant in all these data with males reporting higher rates than females except in Sweden, where females drank more [32]. Illicit drug use was also most widely reported by Czech adolescents [32]. The mean age for sexual debut was 16 years in Sweden [5], 16.7 years in the Czech Republic [33], and 17 years in Germany [34], but similar data for Belgium were not available. Sexual problems attributed to one’s own alcohol use with regards to regretted sex and unprotected sex were reported by 8% of adolescents [32] and again highest levels were found among Czech adolescents (16%). Birth rates per teenage mothers are sometimes used as an indicator for risky sexual behaviour: they were highest in the Czech Republic (14 per 1000 15–19 year old females), followed by Germany (12), and Sweden (7) [35]; Belgian data were not available. Prevalence rates of sexual violence since the age of 15 in females across EU countries showed that the numbers for partner sexual violence ranged from 7% in the Czech Republic to 10% in Sweden, sexual violence by non-partners ranged from 4% in the Czech Republic to 12% in Sweden [36]. Overall, the data do not show a consistent pattern of sexual behaviour in the four countries of interest.

### 1.5. Research Needs

In summary, previous research has shown that just as with risky substance use, sexual risks are a public health concern. For a number of theoretical and empirical considerations, there is reason to believe that substance use and sexual risk are more likely to co-occur. This is because substance use may play a role in sexual coercion and substance use may impair executive control. In addition, common factors are thought to link substance use and sexual risk. Indeed, there are higher rates of sexual risks among high-risk substance users (lifetime and event-level studies). Although some attempts are already being made to take a joint approach to preventive interventions for young people’s alcohol and illicit drug use and sexual risks [37,38], very few preventive interventions simultaneously address both risks. If the risks are indeed overlapping, consideration should be given to addressing both health concerns in one preventive intervention. Data from the current study may help to assess the relevance of this approach in adolescent samples. 

There is inconsistent evidence from college samples about gender differences in risky sexual behaviour, but even less is known about adolescent samples. For sexual victimization, previous research with adolescents consistently shows gender differences. Evidence from college samples suggests that the male/female ratio of reports of sexual victimization may change when reports focus on incidents of sexual victimization with concurrent alcohol and/or drug use, with an increased relative risk for males. This may indicate that gender patterns of sexual victimization may change when substance use is co-occurring. In order to gain an understanding of the mechanisms for male and female adolescent sexual victimization, knowledge about moderators or mediators of sexual victimization are of great importance [19].

Moreover, certain patterns of use increase the likelihood of reporting sexual risks, namely binge drinking and illicit drug use. Both patterns are of interest because they tend to impair situational executive functioning. This may be important because it could help to shape prevention messages to the target group. In addition, previous research has shown that recent substance patterns are associated with a heightened likelihood of sexual risk and that gender may be a moderator of this association, with males being at greater risk. This is also noteworthy because much research has focused on female sexual victimization, e.g. [39], and more insight into understanding male sexual victimization is desirable. 

Finally, alcohol and illicit drug use and sexual behaviour are diverse phenomena among young people in European countries [40], which merits a country-specific approach. As many prevention efforts are designed at the national level, we believe that country-specific reporting of the present research is important to inform stakeholders in each country.

This study is based on adolescents with a history of risky alcohol and/or illicit drug use. We hypothesize that sexual risky behaviour is evenly distributed among young men and women in this group but assume higher levels of sexualized touching and assault victimization in women. Specific research questions are as follows: How high are levels of sexual risks while being drunk and/or high on drugs in adolescents and are there any gender- or country-related differences in proportions of the named sexual risks? Is there an association between sexual risks and past 30-day binge drinking or past 30-day illicit drug use? Do gender or country moderate the association? 

## 2. Material and Methods

### 2.1. Design

Participants of the study were drawn from a randomized-controlled trial assessing the effects of a web-based screening and brief intervention for adolescents with risky alcohol and/or drug use in Sweden, the Czech Republic, Flemish-speaking Belgium, and Germany. The current study is a secondary analysis of this trial. The participating countries were not specifically selected to take part in the trial, but resulted from the fact that they hosted research groups with an interest in developing digital interventions for risky substance use among young people and joined a respective European research initiative. Yet, the participating countries showed substance use levels, which seemed particularly concerning: the Czech Republic, Belgium, and Germany ranked first, third, and sixth, respectively, among 38 European and North American countries for alcohol use among 10th graders [32]. Sweden stands out in terms of levels of single occasion alcohol use, which was particularly high among 10th graders with last day alcohol use [32]. Participants aged from 16 to 18 years were invited to take part at the study by offering them to learn more about adolescent and young adult alcohol and illicit drug use in an individualized format. Recruitment was facilitated by promoting the project’s landing page through flyer cards distributed in schools, cafés, youth clubs, and youth-specific events, and online through social media, advertisements, or affiliated websites. Online data assessments were conducted simultaneously between June 2011 and March 2012.

To secure confidentiality, participants were asked to log on the portal with an anonymous user name. Thereafter, participants gave informed consent to study participation. They were then invited to complete the CRAFFT, a common tool for screening at-risk use of alcohol and/or drugs (“Do you ever use…” alcohol and/or other drugs while riding a Car, to Relax, while Alone, to Forget, despite concerned Friends/Family or causing Trouble?; (CRAFFT); [41]). Participants who scored positive on more than one item of the CRAFFT were invited for study participation. For the current analysis, we only used cross-sectional baseline data of study participants who finalized the assessment. For details regarding the brief intervention trial, see Arnaud and colleagues [42]. The trial was publicly registered and ethical approval was obtained from Regionala Etikprövningsnämnden Stockholm, Ethik-Kommission der Ärztekammer Hamburg, Universiteit Antwerpen (Comite voor Medische Ethiek), and Etická Komise Psychiatrického Centra Praha.

### 2.2. Measures

#### 2.2.1. Sociodemographics

Gender (0 = female, 1 = male), age (“How old are you?” age in years), country (Sweden, Germany, Belgium, Czech Republic), participants’ current school attendance (yes/no), parental educational attainment (“What best describes your mother’s/father’s educational level?” low/middle/high), age at first alcohol use (“At what age (if ever) have you first had an alcoholic drink?” never/age in years) and number of intoxications in the past 30 days (“During the last 30 days, how many times have you gotten drunk or very high from alcohol?” 0–30) were assessed.

#### 2.2.2. Risky Sexual Behaviour

We defined risky sexual behaviour as incidents, in which participants report having sex while being high on drugs and/or alcohol (“Have you ever had sex while being high on drugs and/or alcohol?” yes/no), and not having used a condom (“If yes, did you use a condom?” yes/no). Our questions were modelled after items previously used by Seth and colleagues [9].

#### 2.2.3. Reported Sexualized Touching

We used an item that had been used in previous studies [15] for reported sexualized touching victimization while being drunk and/or high on drugs (“Have you ever been pawed or forced to touch somebody in a sexual way while being high on alcohol and/or drugs?” yes/no).

#### 2.2.4. Reported Sexual Assault

Lifetime experience of reported sexual assault while being high on drugs and/or alcohol was indicated by the question “Have you ever felt forced to have sex while being high on drugs and/or alcohol?” (yes/no). The question was modelled after the item used in the study by Landstedt and Gådin [15].

#### 2.2.5. Binge Drinking

The Alcohol Use Disorder Identification Test (AUDIT-C; [43]) was used to assess past 30-day drinking habits. Of all AUDIT-C items, we were especially interested in acute effects of alcohol, which might impair executive functioning, so we decided to focus on the item relating to binge drinking of the AUDIT-C (“How often did you have five (four for girls) or more drinks on one occasion, like during a party or on one night?” 0 “never”, 1 “once”, 2 “2–4 times a month”, 3 “2–3 times a week”, and 4 “four or more times a week”).

#### 2.2.6. Illicit Drug Use

Illicit drug use in the past 30 days was assessed using the Drug Use Disorder Identification Test (DUDIT; [44]). In the current analyses, we focus on any drug use (“How often did you use drugs other than alcohol?” 0 “never”, 1 “once”, 2 “2–4 times a month”, 3 “2–3 times a week”, and 4 “four or more times a week”). At the time of data collection, the legal situation in all participating countries included cannabis as an illicit drug.

### 2.3. Statistical Analyses

Descriptive statistics were used to characterize participant demographics. Missing value analyses showed that missing answers on reports on sexual risks, past 30-day binge drinking, and past 30-day illicit drug use were not related to any other used variable in a meaningful way. Proportions of participants with sexual risks were reported both for the total sample and separately for countries and gender. Aggregated proportions were calculated using both unweighted and weighted data, thereby accounting for imbalances in gender or country ratio. χ^2^- and *t*-tests were calculated to compare country and gender groups. We calculated effect sizes for weighted proportions (Cohen’s h) and use Cohen’s [45] classification of effect size to interpret results. To assess the association between past 30-day binge drinking and past 30-day illicit drug use, and sexual risks we performed logistic regression analyses to explore whether sexual risks can be predicted by binge drinking or illicit drug use when controlling for country and gender influences. In a first step, we calculated logistic regressions predicting sexual risks by gender and country using Sweden as reference. In a second step, we added either binge drinking or illicit drug use to logistic regressions, thus controlling for gender and country. To check for multicollinearity, we found correlations between predictors ranging between *r* = −0.02 and 0.14 indicating there was no problem. To explore if found associations between sexual risks and substance use differ across countries or gender groups, we extended previous logistic regressions by adding an interaction term between gender and substance use (past 30-day binge drinking or past 30-day illicit drug use) or country and substance use. If the Wald statistic of the interaction term indicated (*p* < 0.05; [46]) that the strength of the relationship between sexual risk and substance use varied across country or gender groups, we added stratified analyses on these associations.

## 3. Results

### 3.1. Sample Characteristics

Data were available from a total of 1449 participants, all of whom reported risky alcohol and/or drug use as defined by the CRAFFT. Participants were unevenly distributed among the four participating countries with the largest proportion coming from the Czech Republic (*n* = 909), followed by Sweden (*n* = 251), Germany (*n* = 146), and Belgium (*n* = 143). Gender proportions were balanced in Belgium (50.3% women) and the Czech Republic (46.0% women); the Swedish subsample had more females (66.5%) and the German subsample fewer female participants (28.0%). The majority of the sample was 16 years old and currently attending school (Table 1). 

A total of 36.9% of the sample reported not having been intoxicated due to substance use in the past 30 days; however, illicit drug use in the past 30 days was reported by over 40% of the sample.

### 3.2. Sexual Risks While Being Drunk and/or High on Drugs

An overview of results is given in Table 2. Almost half of the participants reported having had sex while being drunk and/or high on drugs (49.8%). As hypothesized, the proportion was not significantly different in young women (50.5%) and young men (47.6%; *h* = 0.06) in the total sample. The gender difference varied slightly across different countries, being small in Belgium (*h* = 0.20), similarly directed and even less pronounced in Sweden (*h* = 0.13) and the Czech Republic (*h* = 0.11) and reversed in Germany, where young men more often reported having had sex while being drunk and/or high on drugs than young women (*h* = −0.20). Swedish participants significantly more often reported having had sex while being drunk and/or high on drugs than participants from Belgium (*h* = 0.44 **), the Czech Republic (*h* = 0.40 **), and Germany (*h* = 0.26 *). About half of participants who reported having had sex while being drunk and/or high on drugs additionally reported having had condomless sex on these occasions (48.3%). In line with our hypothesis, no significant gender difference was obtained regarding condom use (*h* = 0.07), although Belgian data showed a small (non-significant) effect (*h* = 0.29), with females appearing to be less cautious than males. German adolescents reported significantly lower levels of condomless sex than other countries (Germany-Sweden: *h* = 0.24**; Germany-Belgium: *h* = −0.32; Germany-Czech Republic: *h* = −0.39 **).

A significant proportion of participants (21.5%) reported having experienced sexualized touching while being drunk and/or high on drugs, and as hypothesized, levels were higher among young women (22.7%) than men (18.1%) across all countries, but the gender difference was not even small (*h* = 0.11 *) according to Cohen [45]. No gender differences were obtained in country subsamples except for Sweden, where young women reported sexualized touching victimization significantly more often (44.5%) than young men (28.8%; *h* = 0.33 **). With regards to country differences, significantly higher levels of reported sexualized touching victimization were reported in Sweden (36.7%) as opposed to other countries with nearly small to large effects (*h* = 0.15 ** – *h* = 0.77 **). Czech respondents reported sexualized touching victimization more often (29.7%) than respondents from Belgium (8.5%; *h* = −0.56 **) and Germany (6.8%; h = 0.63 **). In the total sample, 9.9% reported having experienced sexual assault while being drunk and/or high on drugs. The proportion was significantly higher among young women (11.3%) than men (7.8%), but the overall effect again was not even small (*h* = 0.12 *) except for Swedish data (young Swedish women 21.6%, young Swedish men 12.3%; *h* = 0.25 *). With regards to different countries, the proportion of reported sexual assault victimization among young men and women was over twice as high in Sweden (17.0%) than in any of the other countries, namely Belgium (7.8%; *h* = 0.33 **), the Czech Republic (7.9%; *h* = 0.32 **), and Germany (5.6%; *h* = 0.43 **). 

### 3.3. Past 30-Day Binge Drinking, Past 30-Day Illicit Drug Use, and Sexual Risks

Analyses for binge drinking (see Table 3) reveal that higher levels of past 30-days binge drinking were associated with greater odds of having reported having had sex while being drunk and/or high on drugs (adjusted OR = 2.35, 95% CI: 2.00 to 2.75, *p* = 0.000). The same was true for illicit drug use (see Table 4), which also was associated with greater odds of having sex while being drunk and/or high on drugs (adjusted OR = 1.73, 95% CI: 1.53 to 1.94, *p* = 0.000). There was no association between binge drinking and condomless sex (adjusted OR = 1.10, 95% CI: 0.94 to 1.29, *p* = 0.254) or illicit drug use and condomless sex (adjusted OR = 1.03, 95% CI: 0.92 to 1.16, *p* = 0.585). Associations between reported sexualized touching victimization and binge drinking were found (OR = 1.43, 95% CI: 1.22 to 1.67, *p* = 0.000), and associations between sexualized touching victimization and illicit drug use were observable but less pronounced (OR = 1.30, 95% CI: 0.30, *p* = 0.000). Higher levels of past 30-day binge drinking were associated with greater odds of reported sexual assault victimization (OR = 1.62, 95% CI: 1.29 to 2.03, *p* = 0.000). Higher levels of past 30-day illicit drug use were also related to greater odds of reporting sexual assault victimization (OR = 1.43, 95% CI: 1.23 to 1.67, *p* = 0.000). 

Further explorations of the moderating effects of gender or country on relations between binge drinking or illicit drug use and sexual risks were accomplished by including interaction terms (binge drinking x country, binge drinking x gender, illicit drug use x country, illicit drug use x gender) into logistic regression models for sexual risks in a further step. Logistic regression analyses accompanying interaction terms including gender and country showed that it would be necessary to further explore associations with gender, while interaction terms with country were not significant. Specifically, logistic regression models for sexualized touching victimization (Wald statistic = 4.43, *p* = 0.035) and sexual assault victimization (Wald statistic = 12.83, *p* = 0.000) and past 30-days binge drinking indicated associations between binge drinking and named sexual victimization varied across gender groups. We therefore analysed the association between binge drinking and sexual victimization separately for young women and young men (see Table 5) and found associations were present in young men (sexualized touching victimization OR = 1.72, 95% CI: 1.36 to 2.18, *p* = 0.000; sexual assault victimization OR = 2.59, 95% CI: 1.82 to 3.69, *p* = 0.000) but not in young women. For illicit drug use, reporting having sex while being drunk and/or high on drugs seemed to vary across gender (Wald statistic for the interaction term illicit drug use x gender = 14.93, *p* = 0.000) too. This time, the association between illicit drug use and reporting having sex while being drunk and/or high on drugs was stronger in young women (OR = 2.47, 95% CI: 1.97 to 3.11, *p* = 0.000) than in young men (OR = 1.47, 95% CI: 1.27 to 1.69, *p* = 0.000). 

## 4. Discussion

Our analyses of sexual risks show that a considerable number of adolescents with a history of risky substance use report sexual risks while being drunk and/or high on drugs. In addition, we aimed to assess the potential association between sexual risks and adolescents’ levels of past 30-day binge drinking and past 30-day illicit drug use as an additional marker of adolescents’ ongoing and potential habitual substance use beyond the positive CRAFFT screening. As in previous research, past 30-day binge drinking and illicit drug use were of relevance regarding sexual risks while being drunk and/or high on drugs except for condomless sex.

About half of the present sample report having had sex while being drunk and/or high on drugs. This is noteworthy, as a number of European countries have implemented (e.g. Sweden, Spain) strict consent laws to sexual acts. Assuming that under the influence of alcohol this consent cannot be given, this is a noteworthy finding. As hypothesized, the proportion was the same in both gender groups. Participants indicating having had sex while being drunk and/or high on drugs were especially common in Sweden. Participants who report higher levels of binge drinking and higher levels of illicit drug use were more likely to engage in sex while being drunk and/or high on drugs.

Nearly half of participants, who had indicated they have had sex while being drunk and/or high on drugs, reported condomless sex on these occasions. More research is needed, however, to assess how alarming this finding is; some professionals regard any condomless sex as health-compromising, others underline that we do not know whether condomless sex occurred in steady relationships and/or while using other contraceptives. Although a majority of adolescents report making first sexual experiences in steady relationships [34], doubts remain about complete sexual faithfulness, at least when considering data from young adults [47]. Therefore, we believe that the high frequencies of condomless sex need attention. Frequencies of condomless sex were significantly lower in Germany than in any of the other studied countries. While public health campaigns in Germany encourage condom use, we are not aware of any specific German prevention efforts to target condom use while under the influence of alcohol that could explain this effect. Both levels of past 30-day binge drinking and past 30-day illicit drug use were unrelated to condomless sex while being drunk and/or high on drugs. This is somewhat surprising, as we would have expected binge drinking or illicit drug use habits to impair executive functioning and risk perception, and thus have a detrimental effect on condom use. Perhaps condom use is an automatic habit among those who use them and is therefore not further influenced by binge drinking or illicit drug use.

Reports on sexual victimization while being drunk and/or high on drugs were rather frequent in the present sample: overall, a fifth of the present sample reported sexualized touching victimization while being drunk and/or high on drugs and nearly a tenth reported sexual assault victimization while being drunk and/or high on drugs.

Other than hypothesized, the current data do not consistently show that sexualized touching victimization was predominantly experienced by females; while the proportion of affected young men was significantly lower than that of women across all countries, the difference had a very small effect size. An exception was Sweden, where levels of sexualized touching victimization while being drunk and/or high on drugs were markedly higher for both young men and young women. Additionally, a clear gender difference emerged with young Swedish women reporting higher proportions of sexualized touching victimization than young Swedish men. In addition, in Sweden, more young women than young men reported sexual assault victimization and levels in both gender groups were higher than in any other studied country; sexual assault victimization of young women was over twice as high as compared to any other participating country.

Overall, gender differences in levels of sexual victimization were far less pronounced than we had hypothesized and which are reported in studies on sexual assault victimization, which assessed sexual victimization irrespective of concurrent alcohol and/or drug use as for example in studies in the Netherlands [17] or Sweden [15]. The important difference between these two studies and the present study is (1) the positive CRAFFT-screening result for risky substance use in the present sample and (2) our focus on sexual assault victimization while being drunk and/or high on drugs. Future research is needed to find out whether the gender ratio in sexual assault victimization is different among adolescents with risky substance use and/or whether the same is true for sexual assault victimization while being drunk and/or high on drugs.

Similar to the study by Snipes and colleagues [12], gender played a significant role in the prediction of sexualized victimization while being drunk and/or high on drugs by past 30-day binge drinking. In both studied forms of sexualized victimization, young men with higher levels of past 30-days binge drinking had markedly higher odds of reporting sexual victimization, while no such association was found in young women. This effect could be observed across all countries. We assume that the interplay between gender and sexual victimization changes with the addition of alcohol use, and that men are more vulnerable when reporting binge drinking in particular. A ceiling effect may be at work in relation to female sexual victimization, with women being more vulnerable to sexual assault overall, independent of other factors such as substance use. In contrast, men, who are generally less likely to be sexually victimized, may become more vulnerable as their alcohol use increases; reasons for this are difficult to pinpoint from the present research, but may include situational factors in cases of acute intoxication, because of common factors between sexual victimization and substance use, such as spending time with deviant peers or lack of supervision by carers. Alternatively, risky patterns of substance use such as binge drinking may be a coping strategy among those who have previously experienced sexual victimization. Still, too little is known about male sexual victimization and the added risk alcohol use plays therein. More detailed and longitudinal data, which would allow the sequencing of experiences of sexual victimization and substance use, would be of great value here.

Generally, the countries’ proportions of sexual risks differed in parts considerably with at times large effects. This underscores the need to consider cultural and societal characteristics in studies of substance use and sexual behaviour. At the same time, country-related differences were predominantly found in levels of sexual risks, the associations of sexual risks, and past 30-day binge drinking or illicit drug use were not moderated by country. With respect to country-specific differences, the high levels of both sexual assault victimization (17.0%) and sexualized touching victimization (36.7%) while being drunk and/or high on drugs in Sweden stood out. One explanation could be that while the level of substance use among young people in Sweden tends to be generally lower, for example in terms of alcohol use in the last 30 days [32], the average doses of alcohol used on drinking occasions are markedly higher (Sweden: 7 centilitres of 100% alcohol (cl)) than in Germany (5.6 cl), the Czech Republic (5.6 cl), or Belgium (4.7 cl; [32]). It seems that once Swedish adolescents drink, they tend to drink high doses. This may have an impact on the risk of sexual victimization, as alcohol may impair sexual decision-making and there may be a higher risk of encountering sexually aggressive peers whose inhibitions are impaired by high alcohol consumption. Another central question is whether the high Swedish figures reflect higher incidents of sexual victimization or whether they are due to differences in reporting behaviour. In an overall gender equality index, Sweden ranked highest among all EU countries [36]. It is known that women’s victimization disclosure is higher when overall gender equality is high [36]. The present data could be explained when assuming that a higher level of gender equality also influences the reporting behaviour of men, in the sense that they also report sexual victimization more openly.

There are several strengths to the current study. The study’s multinational approach allowed for country-sensitive analyses. While the study was not powered for the current analyses, the sample was large enough in most cases to compare sufficiently large subgroups.

Of course, there are several limitations to the current study. The data presented here were collected in the early 2010s. While we believe this does not affect associations between substance use and sexual risks, patterns of substance use may have shifted in recent years. The lack of a control group of participants without a history of risky substance use limits our understanding to adolescents who reported previous risky substance use. We have not assessed sexual risks “as such” but asked for incidents while “being drunk and/or high on drugs”, a subcategory of sexual risks. We intended to do so, because we did not aim at targeting a potential general association. Our aim was to illuminate the potential of integrating sexual risks into prevention efforts against risky substance use in adolescence and young adults. We felt linking both phenomena would allow a straightforward communication about risky substance use and sexual risks to prevention recipients and professionals. The cross-sectional nature of the present study does not allow inference on sequencing of events or causal mechanisms of risk association. In order to gain a better and more comprehensive understanding of sexual risks, sexual victimization, and alcohol and drug use patterns, longitudinal event-level studies would be of value. Several previous studies on alcohol use and sexual risks have limited their analyses to sexually active participants. We have refrained from such an approach because we wanted to assess the need for integrating sexually oriented prevention efforts into preventive interventions against alcohol and drug use. For an accurate needs assessment, we therefore intended to assess the levels of substance-related risky sexual incidents in a sample of participants taking part in a prevention initiative against risky alcohol and drug use. Still, our study might be prone to selection bias since our sample consists of adolescents who consented to participate in the brief-intervention study and completed its baseline assessment. The fact that a disproportionately high percentage of the sample came from the Czech Republic is another limitation. This can be explained by more intensive offline recruitment in this country and less strict regulations by the education authorities to allow schools to participate in prevention research initiatives, while at the same time requiring schools to cover substance use prevention. A final limitation is that the German sample had a low proportion of female participants. Explanations for this may be related to a greater reluctance of women to use computers in Germany at this time [48], and a generally lower level of alcohol use among females compared to males in Germany compared to Belgium (no gender difference; [32]) or Sweden (more alcohol use among females; [32]), which could then have affected the initial CRAFFT screening. Our results can therefore not be generalized to the total population of adolescents with risky substance use.

## 5. Conclusions

In sum, the present study showed that when targeting adolescents with a history of risky substance use, we also approach a target group with considerable risk for sexual risks. When developing preventive interventions against risky substance use in adolescents, an additional focus should be set on integrating steps against sexual risks. In our view, more should be done to follow this approach. The present study confirms the potential of such a strategy.

## Figures and Tables

**Table 1 ijerph-20-07002-t001:** Sample description—proportions (%) or means (M) and standard deviations (SD) of sample measures.

	Full Sample	Sweden	Germany	Belgium	Czech Republic
Total *n* after CRAFFT screening	*N* = 1449	*n* = 251	*n* = 146	*n* = 143	*n* = 909
Gender female, % (SD)	48.2 (0.50)	66.5 (0.47) ^A,B,C^	28.0 (0.45) ^A,D,E^	50.3 (0.50) ^B,D^	46.0 (0.50) ^C,E^
Age, M (SD)	16.8 (0.74)	16.8 (0.69)	16.8 (0.80)	16.7 (0.64) ^A^	16.9 (0.76) ^A^
Currently attending school, % (SD)	97.6 (0.15)	99.2 (0.09)	96.6 (0.18)	98.6 (0.12)	97.1 (0.17)
Fathers’ educational level, % (SD)					
Low	10.5 (0.31)	19.0 (0.39)	5.5 (0.23)	8.3 (0.28)	9.2 (0.29)
Middle	63.9 (0.48)	39.8 (0.49)	48.4 (0.50)	44.6 (0.50)	75.4 (0.43)
High	25.2 (0.43)	41.1 (0.49)	46.1 (0.50)	47.1 (0.50)	15.4 (0.36)
Mothers’ educational level, % (SD)					
Low	7.6 (0.26)	9.3 (0.29)	7.1 (0.26)	9.2 (0.29)	7.0 (0.26)
Middle	66.4 (0.47)	37.6 (0.48) ^A,B^	54.8 (0.50) ^A,C^	44.5 (0.50) ^D^	79.0 (0.41) ^B,C,D^
High	25.9 (0.44)	53.2 (0.50) ^A,B^	38.1 (0.49) ^A,C^	46.2 (0.50) ^D^	13.9 (0.35) ^B,C,D^
Substance use related risk (CRAFFT sum score), M (SD)	2.7 (1.38)	3.0 (1.47) ^a,b,C^	2.7 (1.38) ^a^	2.7 (1.29) ^b^	2.7 (1.36) ^C^
Age at first alcohol use, M (SD)	13.0 (2.25)	13.0 (2.01) ^A^	13.9 (2.00) ^A,B,C^	12.9 (2.11) ^B^	12.8 (2.34) ^C^
Not intoxicated in past 30 days, % (SD)	36.9 (0.48)	27.1 (0.44) ^A,B^	34.2 (0.47) ^c^	46.2 (0.50) ^A,c^	38.6 (0.49) ^B^
Illegal drug use past 30 days, % (SD)	44.0 (0.50)	41.1 (0.49) ^A^	42.4 (0.49) ^A^	60.0 (0.49) ^A,B,C^	42.6 (0.49) ^C^
Number of intoxications last 30 days, M (SD)	2.5 (4.18)	2.7 (3.18) ^A^	2.3 (3.02)	1.8 (3.27) ^A,b^	2.6 (4.67) ^b^

Note. Letters indicate pairs of groups that are significantly different in pairwise tests; α-levels for capital letters *p* < 0.01, for small letters *p* < 0.05. M—mean; SD—standard deviation.

**Table 2 ijerph-20-07002-t002:** Proportions of participants reporting sexual risks—total and by gender/country.

	Proportions for Countries						Proportions for Gender Groups					
	Total Country				Effect Size for Total Country Difference		Young Men			Young Women		Effect Size for Gender Difference
	*n*	%	*SD*	Weighted %	Dyad	*h*	*n*	%	*SD*	*n*	%	*h*
Had sex while being drunk and/or high on drugs												
Sweden	238	63.9	0.48	62.7	SWE-GER	0.26 *	74	59.5	0.49	164	65.9	0.13
Germany	143	51.7	0.50	49.8	SWE-BEL	0.44 **	102	54.9	0.50	38	44.7	−0.20
Belgium	132	40.9	0.49	40.8	SWE-CZR	0.40 **	64	35.9	0.48	68	45.6	0.20
Czech Republic	836	42.8	0.49	42.8	GER-BEL	0.18	448	40.0	0.49	379	45.6	0.11
					GER-CZR	0.14 *						
					BEL-CZR	−0.04						
Total	1349	47.3	0.50				688	43.9	0.50	649	50.7	
Total (weighted)		49.8						47.6			50.5	0.06
Condomless sex a												
Sweden	176	50.0	0.50	46.7	SWE-GER	0.24 *	55	45.5	0.50	121	47.9	0.05
Germany	99	37.4	0.48	35.0	SWE-BEL	−0.08	76	38.2	0.49	22	31.8	−0.13
Belgium	68	51.5	0.50	51.0	SWE-CZR	−0.15 *	32	43.8	0.50	36	58.3	0.29
Czech Republic	561	54.4	0.50	54.5	GER-BEL	−0.32 *	295	52.9	0.50	259	56.0	0.06
					GER-CZR	−0.39 **						
					BEL-CZR	−0.07						
Total	904	51.4	0.50				458	48.9	0.50	438	52.7	0.07
Total (weighted)		48.3						45.1			48.5	
Reported sexualized touching victimization while being drunk and/or high on drugs												
Sweden	237	39.7	0.49	36.7	SWE-GER	0.77 **	73	28.8	0.46	164	44.5	0.33 *
Germany	139	7.9	0.27	6.8	SWE-BEL	0.71 **	101	7.9	0.27	35	5.7	−0.09
Belgium	129	8.5	0.28	8.5	SWE-CZR	0.15 **	62	6.5	0.25	67	10.4	0.14
Czech Republic	820	29.9	0.46	29.7	GER-BEL	−0.06	436	29.1	0.46	414	30.3	0.03
					GER-CZR	−0.63 **						
					BEL-CZR	−0.56 **						
Total	1325	27.3	0.45				672	23.8	0.43	680	30.5	0.11 *
Total (weighted)		21.5						18.1			22.7	
Reported sexual assault victimization while being drunk and/or high on drugs												
Sweden	235	18.7	0.39		SWE-GER	0.43 **	73	12.3	0.33	162	21.6	0.25
Germany	134	5.2	0.22		SWE-BEL	0.33 *	99	5.1	0.22	33	6.1	0.04
Belgium	129	7.8	0.27		SWE-CZR	0.32 **	62	6.5	0.25	67	9.0	0.09
Czech Republic	812	8.0	0.27		GER-BEL	−0.11	432	7.4	0.26	372	8.3	0.03
					GER-CZR	−0.11						
					BEL-CZR	−0.01						
Total	1310	9.6	0.29				666	7.5	0.26	634	11.7	
Total (weighted)		9.9						7.8			11.3	0.12 *

Note. Several participants have missing data for gender. All Cohen’s h effect sizes were calculated with weighted data. ^a^ Only for participants who indicated they had sex while being drunk and/or high on drugs in prior question. SWE—Sweden; GER—Germany; BEL—Belgium; CZR—Czech Republic; * *p* < 0.05, ** *p* < 0.01.

**Table 3 ijerph-20-07002-t003:** Logistic regressions predicting sexual risks by gender and binge drinking in total sample.

Predicted Outcome	Predictors	*B (SE)*	*OR*	95% CI	*p*	Model Parameters
Had sex while being drunk and/or high on drugs	Constant	0.63 (0.14)				χ^2^ = 40.39, df = 4, *p* = 0.000
Model 1 (*n* = 1337)	Gender	−0.20 (0.11)	0.82	0.66–1.03	0.081	Nagelkerke’s R^2^ = 0.04
	Country					−2 Log-Likelihood = 1808.875
	Germany	−0.40 (0.22)	0.67	0.43–1.03	0.070	
	Belgium	−0.91 (0.22)	0.41	0.26–0.63	0.000	
	Czech Republic	−0.83 (0.15)	0.44	0.32–0.59	0.000	
Had sex while being drunk and/or high on drugs	Constant	−0.63 (0.20)				χ^2^ = 169.25, df = 5, *p* = 0.000
Model 2 (*n* = 1124)	Gender	−0.47 (0.13)	0.62	0.48–0.81	0.000	Nagelkerke’s R^2^ = 0.19
	Country					−2 Log-Likelihood = 1388.936
	Germany	−0.13 (0.27)	0.88	0.53–1.48	0.635	
	Belgium	−0.86 (0.26)	0.43	0.26–0.71	0.001	
	Czech Republic	−0.83 (0.17)	0.44	0.31–0.61	0.000	
	**Binge drinking**	0.85 (0.08)	**2.35**	**2.00–2.75**	**0.000**	
Condomless sex ^a^	Constant	−0.02 (0.16)				χ^2^ = 10.76, df = 4, *p* = 0.029
Model 1 (*n* = 896)	Gender	0.06 (0.14)	1.06	0.81–1.40	0.660	Nagelkerke’s R = 0.02
	Country					−2 Log-Likelihood = 1230.722
	Germany	0.52 (0.27)	1.67	0.99–2.82	0.053	
	Belgium	−0.07 (0.29)	0.93	0.53–1.64	0.811	
	Czech Republic	−0.19 (0.18)	0.83	0.59–1.17	0.287	
Condomless sex ^a^	**Constant**	−0.18 (0.22)				χ^2^ = 15.25, df = 5, *p* = 0.009
Model 2 (*n* = 780)	Gender	−0.56 (0.15)	0.95	0.71–1.27	0.705	Nagelkerke’s R = 0.03
	Country					−2 Log-Likelihood = 1065.442
	Germany	0.77 (0.29)	2.17	1.22–3.84	0.008	
	Belgium	−0.04 (0.31)	0.96	0.53–1.76	0.899	
	Czech Republic	−0.16 (0.19)	0.86	0.60–1.23	0.396	
	**Binge drinking**	0.09 (0.08)	**1.10**	**0.94–1.29**	**0.254**	
Reported sexualized touching victimization while being drunk and/or high on drugs	Constant	−0.36 (0.14)				χ^2^ = 84.73, df = 4, *p* = 0.000
	Gender	−0.19 (0.13)	0.83	0.64–1.07	0.148	Nagelkerke’s R^2^ = 0.09
Model 1 (*n* = 1314)	Country					−2 Log-Likelihood = 1450.495
	Germany	−2.03 (0.36)	0.13	0.07–0.26	0.000	
	Belgium	−1.92 (0.34)	0.15	0.08–0.29	0.000	
	Czech Republic	−0.40 (0.16)	0.67	0.49–0.91	0.010	
Reported sexualized touching victimization while being drunk and/or high on drugs	Constant	−0.91 (0.20)				χ^2^ = 97.14, df = 5, *p* = 0.000
	Gender	−0.29 (0.14)	0.75	0.57–0.99	0.045	Nagelkerke’s R^2^ = 0.12
Model 2 (*n* = 1107)	Country					−2 Log-Likelihood = 1219.492
	Germany	−2.55 (0.49)	0.08	0.30–0.20	0.000	
	Belgium	−1.74 (0.35)	0.18	0.09–0.35	0.000	
	Czech Republic	−0.40 (0.17)	0.67	0.48–0.93	0.017	
	**Binge drinking**	0.36 (0.08)	**1.43**	**1.22–1.67**	**0.000**	
Reported sexualized assault victimization while being drunk and/or high on drugs	Constant	−1.38 (0.18)				χ^2^ = 27.29, df = 4, *p* = 0.000
	Gender	−0.30 (0.20)	0.74	0.50–1.10	0.002	Nagelkerke’s R^2^ = 0.04
Model 1 (*n* = 1300)	Country					−2 Log-Likelihood = 791.252
	Germany	−1.29 (0.43)	0.28	0.12–0.64	0.003	
	Belgium	−0.96 (0.37)	0.38	0.19–0.79	0.010	
	Czech Republic	−0.93 (0.22)	0.39	0.26–0.60	0.000	
Reported sexualized assault victimization while being drunk and/or high on drugs	Constant	−2.16 (0.28)				χ^2^ = 41.33, df = 5, *p* = 0.000
	Gender	−0.47 (0.21)	0.62	0.41–0.95	0.026	Nagelkerke’s R^2^ = 0.08
Model 2 (*n* = 1096)	Country					−2 Log-Likelihood = 690.396
	Germany	−1.25 (0.47)	0.29	0.12–0.72	0.008	
	Belgium	−0.79 (0.38)	0.46	0.22–0.97	0.040	
	Czech Republic	−0.90 (0.23)	0.41	0.26–0.64	0.000	
	**Binge drinking**	0.48 (0.12)	**1.62**	**1.29–2.03**	**0.000**	

Note. Reference country is Sweden. Gender: 0 = young women, 1 = young men. ^a^ Only for participants who indicated they had sex while being drunk and/or high on drugs in prior question.

**Table 4 ijerph-20-07002-t004:** Logistic regressions predicting sexual risks by gender, country, and illicit drug use in total sample.

Predicted Outcome	Predictors	*B (SE)*	*OR*	95% CI	*p*	Model Parameters
Had sex while being drunk and/or high on drugs	Constant	0.63 (0.14)				χ^2^ = 40.39, df = 4, *p* = 0.000
Model 1 (*n* = 1337)	Gender	−0.20 (0.11)	0.82	0.66–1.03	0.081	Nagelkerke’s R^2^ = 0.04
	Country					−2 Log-Likelihood = 1808.875
	Germany	−0.40 (0.22)	0.67	0.43–1.03	0.070	
	Belgium	−0.91 (0.22)	0.41	0.26–0.63	0.000	
	Czech Republic	−0.83 (0.15)	0.44	0.32–0.59	0.000	
Had sex while being drunk and/or high on drugs	Constant	0.41 (0.15)				χ^2^ = 131.96, df = 5, *p* = 0.000
Model 2 (*n* = 1212)	Gender	−0.41 (0.13)	0.66	0.52–0.85	0.001	Nagelkerke’s R^2^ = 0.14
	Country					−2 Log-Likelihood = 1548.106
	Germany	−0.41 (0.24)	0.66	0.41–1.06	0.087	
	Belgium	−1.20 (0.25)	0.30	0.19–0.49	0.000	
	Czech Republic	−0.80 (0.17)	0.45	0.32–0.62	0.000	
	**Illicit drug use**	0.55 (0.60)	**1.73**	**1.53–1.94**	**0.000**	
Condomless sex ^a^	Constant	−0.02 (0.16)				χ^2^ = 10.76, df = 4, *p* = 0.029
Model 1 (*n* = 896)	Gender	0.06 (0.14)	1.06	0.81–1.40	0.660	Nagelkerke’s R = 0.02
	Country					−2 Log-Likelihood = 1230.722
	Germany	0.52 (0.27)	1.67	0.99–2.82	0.053	
	Belgium	−0.07 (0.29)	0.93	0.53–1.64	0.811	
	Czech Republic	−0.19 (0.18)	0.83	0.59–1.17	0.287	
Condomless sex ^a^	Constant	0.01 (0.17)				χ^2^ = 8.74, df = 5, *p* = 0.120
Model 2 (*n* = 840)	Gender	−0.04 (0.14)	0.96	0.72–1.27	0.778	Nagelkerke’s R^2^ = 0.01
	Country					−2 Log-Likelihood = 1155.176
	Germany	0.47 (0.27)	1.60	0.94–2.71	0.084	
	Belgium	−0.14 (0.30)	0.87	0.48–1.57	0.646	
	Czech Republic	−0.18 (0.18)	0.83	0.59–1.19	0.312	
	**Illicit drug use**	0.03 (0.06)	**1.03**	**0.92–1.16**	**0.585**	
Reported sexualized touching victimization while being drunk and/or high on drugs	Constant	−0.36 (0.14)				χ^2^ = 84.73, df = 4, *p* = 0.000
	Gender	−0.19 (0.13)	0.83	0.64–1.07	0.148	Nagelkerke’s R^2^ = 0.09
Model 1 (*n* = 1314)	Country					−2 Log-Likelihood = 1450.495
	Germany	−2.03 (0.36)	0.13	0.07–0.26	0.000	
	Belgium	−1.92 (0.34)	0.15	0.08–0.29	0.000	
	Czech Republic	−0.40 (0.16)	0.67	0.49–0.91	0.010	
Reported sexualized touching victimization while being drunk and/or high on drugs	Constant	−0.53 (0.15)				χ^2^ = 97.20, df = 5, *p* = 0.000
	Gender	−0.25 (0.14)	0.78	0.59–1.02	0.065	Nagelkerke’s R^2^ = 0.11
Model 2 (*n* = 1195)	Country					−2 Log-Likelihood = 1322.594
	Germany	−2.14 (0.38)	0.12	0.06–0.26	0.000	
	Belgium	−1.97 (0.35)	0.14	0.07–0.28	0.000	
	Czech Republic	−0.35 (0.16)	0.71	0.51–0.97	0.032	
	**Illicit drug use**	0.26 (0.06)	**1.30**	**1.16–1.45**	**0.000**	
Reported sexualized assault victimization while being drunk and/or high on drugs	Constant	−1.38 (0.18)				χ^2^ = 27.29, df = 4, *p* = 0.000
	Gender	−0.30 (0.20)	0.74	0.50–1.10	0.002	Nagelkerke’s R^2^ = 0.04
Model 1 (*n* = 1300)	Country					−2 Log-Likelihood = 791.252
	Germany	−1.29 (0.43)	0.28	0.12–0.64	0.003	
	Belgium	−0.96 (0.37)	0.38	0.19–0.79	0.010	
	Czech Republic	−0.93 (0.22)	0.39	0.26–0.60	0.000	
Reported sexualized assault victimization while being drunk and/or high on drugs	Constant	−1.68 (0.20)				χ^2^ = 43.89, df = 5, *p* = 0.000
	Gender	−0.39 (0.21)	0.68	0.45–1.02	0.062	Nagelkerke’s R^2^ = 0.08
Model 2 (*n* = 1182)	Country					−2 Log-Likelihood = 719.321
	Germany	−1.45 (0.47)	0.24	0.09–0.59	0.002	
	Belgium	−1.17 (0.40)	0.31	0.14–0.68	0.003	
	Czech Republic	−0.86 (0.23)	0.42	0.27–0.66	0.000	
	**Illicit drug use**	0.36 (0.08)	**1.43**	**1.23–1.67**	**0.000**	

Note. Reference country is Sweden. Gender: 0 = young women, 1 = young men. ^a^ Only for participants who indicated they had sex while being drunk and/or high on drugs in prior question.

**Table 5 ijerph-20-07002-t005:** Adjusted odds ratios for experiencing sexual risks stratified by gender.

Predicted Outcome	Predictor	Gender Group	Adjusted *OR*	95% CI	
				*LL*	*UL*
Sexualized touching victimization	Binge drinking	Female (*n* = 550)	1.21	0.97	1.51
		Male (*n* = 557)	1.72	1.36	2.18
Sexualized assault victimization	Binge drinking	Female (*n* = 543)	1.13	0.84	1.52
		Male (*n* = 553)	2.59	1.82	3.69
Had sex while being drunk and/or high on drugs	Illicit drug use	Female (*n* = 591)	2.47	1.97	3.11
		Male (*n* = 621)	1.47	1.27	1.69

Note. Odds ratios are adjusted for country. CI—confidence interval; *LL*—lower limit; *UL*—upper limit.

## Data Availability

The data presented in this study are available on request from the corresponding author.

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
