# Peer review of "Risky Sexual Behaviour and Sexual Victimization among Young People with Risky Substance Use in Europe—Perspectives from Belgium, Sweden, the Czech Republic, and Germany"

_ijerph, 2023, doi:10.3390/ijerph20217002_

Round 1
Reviewer 1 Report (Previous Reviewer 1)
Comments and Suggestions for Authors
I think the paper is much improved and was a pleasure to read. My only comment is that there seems to be some inconsistency in results. In the results section, you write that the association between sexual victimization and binge drinking was stronger among boys, while the association between illicit drug use and sexual victimization was higher among girls but in the discussion you say that the associations were stronger among boys for both binge drinking and sexual victimization. From the table it appears that the results section is correct. Therefore, something needs to be added in the discussion for why this different pattern may occur.
Comments on the Quality of English LanguageThe English is generally fine, but there are a couple of paragraphs that seem like they should be split into two, in particular the paragraph in section 1.2 and the paragraph starting on line 479. On line 261 you write "We used an item, had been used in previous studies." It appears to be missing a word which should replace the comma. There is a type on 253, od in place of of.
Author Response
Please see the attachment

Reviewer 2 Report (Previous Reviewer 2)
Comments and Suggestions for Authors
I really appreciate the efforts of the authors in their revision process.
It is now easily readable and understandable.
The followings are my few suggestions. Please consider.
page 5 lines #222-223: the statement can be deleted.
page 7 line #310: 'N=' can be deleted or substituted by 'a total of'
Table 2: consider editing since it is really hard to read the contents(especially the numbers) in its current form.
page 18 lines #371-372: I think this statement can be deleted since it is explained in the method section.
page 18 lines #373-376: these statement can be moved to the method section
page 18 lines #410~421: These statements are too elaborative. It would be improved if shortened. consider my suggestion: Logistic regression analyses accompanying interaction terms including gender and country showed that it would be necessary to further explore associations with gender, while interaction terms with country were not significant.
Table 4: Consider merging cells with the same contents in the first and second columns. e.g. merging 2nd and 3rd rows since their contents are the same, 'sexualized touching victimization'
page 24 lines #457-459: may need citations.
page 24 lines 487-491: I think this statement is redundant of the result section so that it can be deleted.
page 26 lines 547-553: These statement might make the potential readers somewhat confusing. I think the manuscript would not be damaged without the statements.
conclusion: I think this manuscript would be a lot more beneficial when its conclusion is revised a bit. I think what is described in the conclusion part of the abstract is better. Using a statement with a citation is not usually seen in a conclusion section.
Comments on the Quality of English Language
page 1 line #90: report --> reported
page 5 line #249: father's and mother's --> can be changed to "parental"
Author Response
Please see the attachment

Reviewer 3 Report (Previous Reviewer 3)
Comments and Suggestions for Authors
In general the authors have been responsive in their review and the manuscript has been improved as a result.
As requested, the authors added the dates of collection for these data. The dates of collection were from June 2011 to March 2012. As these data are around 12 years old, it would be helpful for the authors to note this as a limitation in the discussion section. While the general pattern showing associations between substance use and sexual risk are likely to be unchanged today, the pattern of use of substances may have changed in the past 12 years.
Author Response
Please see the attachment

Reviewer 4 Report (Previous Reviewer 4)
Comments and Suggestions for Authors
Thanks for having the opportunbity to review this revised manuscript. I still have some concerns and suggestions:
(1) it's obvious that there exists co-occurrence of substance use and sexual risks, as they were both common risk or problems among adolescents and greatly influenced by the similar factors. And you also suggest that "Evidence suggests an association between participants’ levels of alcohol and illicit drug use and risky sexual behaviour ". How could this be a innovation and research gap of this research?
(2) 1,449 adolescents is indid relatively a small sample for your research question, especially they are from four countriesï¼›
(3) I am also concerned about the representativeness of the sample, as the the participants were recruited from a web-based intervention against alcohol and illicit drug use;
(4) the gender fifferences and country differences were not clearly discussed in the introduction;
(5) the research data is old -"Online data assessment was conducted between June 2011 and March 2012";
(6) the analysis should be reorginized to clealry present the main points, and the analyis semms to be simple;
Round 2
Reviewer 4 Report (Previous Reviewer 4)
Comments and Suggestions for Authors
I appreciate it very much for the careful response and revision. However, the major concerns of the research (the simiple research question and analysis, as well as the limitations of the sample and the significance of the study) were not well addressed, and all of these concerns could not be improved through revision due to the flaws ofresearch design.
Author Response
Dear reviewer 4,
thank you very much again for your feedback. We would still very much like to convince you of the quality of our manuscript. As for some of the remarks you made, we would like to answer as follows:
You stated in resubmission round 2: “I appreciate it very much for the careful response and revision. However, the major concerns of the research (the simiple research question and analysis, as well as the limitations of the sample and the significance of the study) were not well addressed, and all of these concerns could not be improved through revision due to the flaws ofresearch design.”
We would like to answer as follows: We feel the issue of simplicity, which you had addressed here and in previous feedback sometimes refers to the design, later to the analyses. Now you extend this feedback to the simplicity of the research question (a new point) and the analyses. We would like to counter with our previous argument: whatever is “simple” or “complex” may be disputed. What is important is the relevance of the research question and the adequacy of research methods. We also argue that following the principle of parsimony, the use of less complex statistical analyses can be preferred to more complex ones, as long as the research questions can be answered with the analyses.
With regards to the limitations of the sample, which you address, we would like to remark the following:
The population focused on with this research is the population studied: young people with risky substance use who participate in an intervention against risky substance use. It is no shortcoming of the study, that the sample was made up of participants of a web-based intervention against substance use, because this is the population about whom we want to gain knowledge. This research is not about estimating prevalence rates of sexual risks in a general population of adolescents, for which your argument would have been valid. The only restriction regarding representativeness is mentioned in the manuscript’s discussion: The sample is restricted to participants, who are willing to participate in research on named interventions. Yet, this restriction applies to all research on human participants because giving informed consent is an ethical necessity and a prerequisite for publishing with IJERPH.
As for your remarks on the significance of the research design, we would like to comment:
As to devising research designs with actual mechanisms in the area – we believe we are familiar with the research in the area and we know of virtually no study, that delves into mechanisms of, e. g. sexual victimization. Researchers are figuratively gnashing their teeth out to disentangle the complex interplay between personal factors, situational factors, possible victims, possible perpetrators, context variables and the role age may play therein. Stating that current research is not valuable because it does not identify mechanisms in this area does not, in our view, reflect where researchers are in this area of study.
Thank you again for your consideration of our argumentation. The submitted revised version now also contains issues, another Academic Editor made with regards to the manuscript. They refer to the selection of countries for the trial, the disproportionate large subsample from the Czech Republic and the low proportion of females in the German sample.
Best regards,
Christiane Baldus and colleagues
This manuscript is a resubmission of an earlier submission. The following is a list of the peer review reports and author responses from that submission.
Round 1
Reviewer 1 Report
Comments and Suggestions for Authors
This is a solid piece of work examining the relationships between risky alcohol and drug use and sexual risk. The four country comparison is a significant contribution. In general, the manuscript could use some editing for English grammar, mostly in the incorrect use of prepositions. "Reporting of..." for example, and "explore on" are frequently used.
One area of clarification is needed in the methods. All participants screened as having a high probability of risky drug or alcohol use but over 36.9% reported not have been intoxicated in the last 30 days. This seems rather low. What is the time frame for the CRAFFT questions?
Author Response
Please check with the attachment.

Reviewer 2 Report
Comments and Suggestions for Authors
Thank you for giving me this chance of reviewing the manuscript entitled "Risky sexual behavior and sexual victimization among young people with risky substance use in Europe - perspectives from Belgium, Sweden, the Czech Republic and Germany"
My first impression of the manuscript is that it should be organized in a more scientific way. I don't know how to put it in a simple statement, but the manuscript is too elaborative in a way it misses what it really wanted to focus on.
INTRODUCTION
- The introduction is too long. Try to summarize. In addition, I don't quite get the study objective. Be specific and concise.
- it seems some parts of the introduction should be moved to the methods section.
-line 197: does 'rates of sexual risks' refer to 'proportion of participants reported sexual experience/sexual intercourse'?
MATERIALS AND METHODS
- What was the total number of the study participants intended to recruit?
- The online survey was conducted from when to when?
- line 221~222: As far as I am concerned, informed consent should not only be obtained from the study participants(adolescents) themselves but from their guidance as well. The authors should re-check the ethical process.
- It seems CRAFFT refers to "riding a Car, to Relax, while Alone, to Forget, despite concerned Friends/Family or causing Trouble" . Then, the abbreviation should follow right after the full-term in parentheses.
- from line 235~242: why is it in italics?, how come the educational level of father was not considered as a variable?
- from line 264~266 & 269~271: How come the authors did not consider 1~3 times a week?
- The description for statistical analyses is too long. Try to summarize.
- I think line 288-289 is not necessary.
RESULTS
- line 313: I don't think that it is necessary to italicize "not"
*Table 1
- I don't quite get what the footnote means. Why did the authors use different alphabets for p-values? What groups do the alphabets refer to?
- A footnote for 'M'(mean), SD (standard deviation) is required.
* Table 2
- I think SD is not necessary for table 2.
- the presentation order would be better if presented in the following order; total - male - female (n, %, weighted %)
- To compare differences between countries, it would be better to use the chi-squared test, I think.
*Table 3
- I think the authors should re-think about the presentation style of table 3. I think the format of the supplementary tables are more appropriate. It's really hard to understand what table 3 tries to present in the way it is now.
- If the purpose of this study was to compare risky sexual behaviors of boys and girls in association with alcohol consumption and/or illicit drug use as stated in the introduction section, I believe table 3 is not necessary. The authors should reorganize the manuscript and specifically state the study objective in order to present appropriate tables and results.
DISCUSSION
- Still, I don't get whether the point of this study was on to compare between the countries or on to compare between sexes.
- line 432-433: A discussion for Sweden would be helpful.
- The authors should distinguish 'rates' from 'proportions'
- The authors should distinguish 'gender' from 'sex'
CONCLUSIONS
- The conclusion of the manuscript should be based on the study results. The present conclusion is rather focusing on the preventive health behavior interventions.
REFERENCES
- please check the citation style of the journal(https://www.mdpi.com/journal/ijerph/instructions#preparation).
Author Response
please check with the attachment.

Reviewer 3 Report
Comments and Suggestions for Authors
Risky sexual behaviour and sexual victimization among young people with risky substance use in Europe – perspectives from Belgium, Sweden, the Czech Republic and Germany (IJERPH-2331575)
The manuscript describes data from a cross-sectional study of adolescents/emerging adults (ages 16-18) recruited from Belgium, Sweden, the Czech Republic, and Germany who were screened positive for some aspects of substance use and responded to additional questions assessing substance use and sexual risk behavior. Substantial numbers of participants reported having sex under the influence of substances, unprotected sex, and unwanted sexual touching. Substance use, especially binge drinking was associated with some sexual risk. There were a few differences between countries of origin.
Strengths of the study include the international recruitment, large sample size (N = 1449), and examination of an important topic in a vulnerable population. Limitations include some aspects of the presentation of methods and findings.
Comments below may serve to strengthen the manuscript.
1. The manuscript notes that these data were collected as part of a randomized clinical trial. Were these taken from the baseline? If so, that would be worth mentioning. If taken from some other assessment after the trial started, it would be important to describe how the intervention may have impacted the findings.
2. When was recruitment conducted?
3. For those of us not familiar with the laws in each of these four countries, it would helpful to indicate if illegal drug use, as described in the manuscript, includes the use of cannabis.
4. For the subset of analyses in table 3 that examined condomless sex, the footnote indicates these analyses only included individuals who had reported sex while intoxicated. I was unsure why the analyses were limited to this group since condomless sex could be higher risk even if the individuals are sober. What was the rationale for this choice? Also, what was the sample size for these analyses since they did not include everyone?
Author Response
Please check with the attachment.

Reviewer 4 Report
Comments and Suggestions for Authors
Though thi is is an interesting theme, some contents were not clearly stated:
(1) the introduction should be expanded, for example, the major fctors accouting for Risky sexual behaviour and sexual victimization should be summarized, so as to strengthen the significance of risky substance use;
(2) the research question was not clearly stated, especially the neessity and contributions of this study; and it seems that the relations among Risky sexual behaviour, sexual victimization and alcohol and/or illicit drug use, had been examined;
(3)this study was conducted in different European countries. though you stated the cultural and societal context differences, how theywere reflected among four different European countries - Sweden, Germany, the Czech Republic and Belgium? this was not presented and discussed in the study;
(4) there was no comments and summarization in the inroduction;
(5) what are the importance to distinguish the past 30-day alcohol and past 30-day illicit drug use and sexual risks?
(6) the research design is simple, and noly examined the direct relation; more important issues (e.g., the mechanism) were not examined, which largely damage the significance of this study;
(7) the discussion should be further expanded to discuss the main findings.
